# TTBK2 and primary cilia are essential for the connectivity and survival of cerebellar Purkinje neurons

Emily Bowie[1], Sarah C Goetz[2]*

[1]University Program in Genetics and Genomics, Duke University, Durham, United States; [2]Department of Pharmacology and Cancer Biology, Duke University, Durham, United States

**Abstract** Primary cilia are vital signaling organelles that extend from most types of cells, including neurons and glia. These structures are essential for development of many tissues and organs; however, their function in adult tissues, particularly neurons in the brain, remains largely unknown. Tau tubulin kinase 2 (TTBK2) is a critical regulator of ciliogenesis, and is also mutated in a hereditary neurodegenerative disorder, spinocerebellar ataxia type 11 (SCA11). Here, we show that conditional knockout of *Ttbk2* in adult mice results in degenerative cerebellar phenotypes that recapitulate aspects of SCA11 including motor coordination deficits and defects to Purkinje cell (PC) integrity. We also find that the *Ttbk2* conditional mutant mice quickly lose cilia throughout the brain. We show that conditional knockout of the key ciliary trafficking gene *Ift88* in adult mice results in nearly identical cerebellar phenotypes to those of the *Ttbk2* knockout, indicating that disruption of ciliary signaling is a key driver of these phenotypes. Our data suggest that primary cilia play an integral role in maintaining the function of PCs in the adult cerebellum and reveal novel insights into mechanisms involved in neurodegeneration.

*For correspondence:
sarah.c.goetz@duke.edu

**Competing interests:** The authors declare that no competing interests exist.

## Introduction

Primary cilia are organelles that serve as compartments that mediate and integrate essential signaling pathways, including Hedgehog (HH) signaling. Because of their critical roles in developmental signaling pathways (*Goetz and Anderson, 2010*), disruptions to cilium assembly, structure, or function are associated with a number of hereditary developmental syndromes, collectively termed ciliopathies. Among the more common pathologies associated with ciliopathy are a variety of neurological deficits (*Reiter and Leroux, 2017*). During development, ciliary signals drive proliferation and patterning of neural progenitor populations (*Guemez-Gamboa et al., 2014*). Cilia then persist on post-mitotic neurons through adulthood (*Sterpka and Chen, 2018*). However, we have only a limited understanding of the roles for primary cilia on differentiated, post-mitotic neurons, particularly within the adult brain.

Newly emerging evidence suggests that cilia and ciliary signaling are important in adult neurons: Cilia are required for the establishment of synaptic connectivity in hippocampal dentate granule neurons (*Kumamoto et al., 2012*) and in striatal interneurons (*Guo et al., 2017*). Neuronal cilia also concentrate a wide array of G-protein coupled receptors (GPCRs) and other neuropeptide and neurotrophin receptors that are important for complex neurological functions (*Berbari et al., 2008*; *Domire et al., 2011*; *Green et al., 2012*; *Guadiana et al., 2016*). Defects in cilia structure have also been observed in patient samples and animal models of several neurodegenerative and neuropsychiatric conditions (*Chakravarthy et al., 2012*; *Dhekne et al., 2018*; *Keryer et al., 2011*; *Muñoz-Estrada et al., 2018*).

**eLife digest** Many mammalian cells have a single hair-like structure, known as the primary cilium that projects away from the surface of the cell. This small projection from the membrane regulates many signaling pathways, particularly during embryonic development. However, most of the neurons in the adult brain also have primary cilia, and it is not yet understood what the role of the primary cilium has in maintaining most adult tissues.

The primary cilium needs the protein TTBK2 to assemble, and mutations in the gene that codes for this protein cause a neurodegenerative disorder that first appears in adulthood known as spinocerebral ataxia type 11 (SCA11). People with this disease have a movement disorder caused by the loss of neurons called Purkinje cells in the cerebellum. In 2018, researchers showed that mutated versions of TTBK2 associated with SCA11 interfere with the role of normal TTBK2 in assembling the cilium. But it was unclear whether primary cilia are required for the survival of Purkinje cells in the cerebellum.

Now, Bowie and Goetz (who are two of the researchers that conducted the 2018 study) have found that deleting the gene that codes for TTBK2 in the brain of adult mice leads to the loss of cilia, followed by impaired movement. Additionally, the connections between Purkinje cells and other neurons are lost, and Purkinje cells eventually degenerate and die. If the cilia are removed using a different mechanism, the results are the same, showing for the first time that primary cilia are important to keep Purkinje cells alive and connected to other neurons.

These results shed light on the roles of primary cilia within adult tissues, and provide insight into the mechanisms underlying SCA11, a neurodegenerative disease for which no treatment currently exists. In the future, it will be important to extend the results of this study to other types of neurons affected in different neurodegenerative conditions. Ultimately, this line of research could lead to uncovering the causes of certain neurodegenerative disorders and provide new paths to treatment.

We previously showed that Tau tubulin kinase 2 (TTBK2), a kinase causally mutated in the hereditary neurodegenerative disorder spinocerebellar ataxia type 11 (SCA11) (*Houlden et al., 2007*), is an essential regulator of ciliogenesis (*Goetz et al., 2012*). These mutations are frameshift-causing indels that result in premature truncation of TTBK2 at ~AA 450. SCA11 is characterized by a loss of Purkinje cells (PC) in the cerebellum, causing ataxia and other motor coordination deficits (*Houlden et al., 2007*; *Seidel et al., 2012*). Recently, we demonstrated that SCA11-associated alleles of *Ttbk2* act as dominant negatives, causing defects in cilium assembly, stability, and function (*Bowie et al., 2018*).

Given the association between the SCA11-associated truncations of TTBK2 and ciliary dysfunction, we set out to test whether loss of TTBK2 function within the adult brain is associated with degeneration of cerebellar neurons. The cerebellum is the region of the brain responsible for controlling motor coordination, learning, and other cognitive functions. The development and morphogenesis of the cerebellum depends on primary cilia, which are critical for expansion of granule neuron progenitors (*Chizhikov et al., 2007*; *Spassky et al., 2008*). PCs, granule neurons, and interneurons, as well as Bergmann glia (BG), are ciliated in the adult cerebellum as well as during development. However, the roles of cilia and ciliary signaling in the adult cerebellum are unknown.

In this study, we show that global conditional knockout of *Ttbk2* during adulthood as well as genetic targeting of cilia using *Ift88* conditional knockout mice, cause similar degenerative changes to cerebellar connectivity. These cellular changes are accompanied by motor coordination phenotypes in the mice. We demonstrate that loss of *Ttbk2* and cilia leads to altered intracellular $Ca^{++}$ in PCs, loss of VGLUT2+ synapses on PC dendrites, and general dysfunction of these cells. We provide strong evidence that primary cilia and ciliary signals are important for maintaining connectivity of specific neurons within the brain, and we demonstrate that dysfunction of primary cilia can cause or contribute to neurodegeneration within the mammalian brain.

## Results

### Loss of *Ttbk2* from the adult brain causes SCA-like cerebellar phenotypes

Mutations within *TTBK2* cause the adult-onset, neurodegenerative disease SCA11. However, the etiology of SCA11 is poorly defined. SCA11 is somewhat unusual among SCAs, in part because the reported causal mutations are base pair insertions or deletions within the coding region of *TTBK2* (*Houlden et al., 2007*; *Johnson et al., 2008*; *Lindquist et al., 2017*), rather than the expansion of CAG repeats, which is the genetic cause of most SCA subtypes (*Hersheson et al., 2012*). To test the requirements for TTBK2 in maintaining neural function within the adult brain, we obtained a conditional allele of *Ttbk2* (*Ttbk2^tm1c(EUCOMM)Hmgu*) from the European Mutant Mouse Cell Repository, (referred to from here as *Ttbk2^fl*). We then crossed *Ttbk2^fl* mice to a mouse line expressing tamoxifen-inducible Cre recombinase driven by a ubiquitously expressed promoter, *Ubc-Cre-ERT2* (*Ruzankina et al., 2007*). Using this model, we induce recombination of *Ttbk2* in all tissues of the mouse, including the brain, upon injection with tamoxifen (TMX). Because morphogenesis of the mouse cerebellum is complete by P21 (*Marzban et al., 2014*), we chose this time to begin our TMX injections. For all of our experiments, Control animals are either siblings with the same genotype (*Ttbk2^fl/fl;Ubc-Cre-ERT2^+*) injected with oil vehicle only, or *Ttbk2^fl/fl;Ubc-Cre-ERT2^-* sibling mice injected with the same dose of TMX. We found no phenotypic differences between Control condition animals or pre-induction *Ttbk2^fl/fl;Ubc-Cre-ERT2+* animals at P21 (*Figure 1—figure supplement 1A-C*). Consistent with other conditional mutants where cilia are globally removed in adulthood (*Davenport et al., 2007*), 4-month-old Ttbk2^c.mut mice exhibit obesity (*Figure 1—figure supplement 1D, D', E*: 32.29 g ± 1.86 for Control vs. 46.33 g ± 2.04 for Ttbk2^c.mut) as well as cystic kidneys (*Figure 1—figure supplement 1F*). Loss of TTBK2 protein was confirmed with western blot analysis on cerebellum lysates from Ttbk2^c.mut animals and littermate Controls (*Figure 1—figure supplement 2G*).

Because the cerebellum is critical for motor coordination and SCA11 is associated with motor deficits, we evaluated locomotor behavior in the *Ttbk2^fl/fl;Ubc-Cre-ERT2^+*,TMX treated animals (referred to from here as Ttbk2^c.mut) relative to littermate Controls. Within 3 weeks following induction of recombination with TMX, Ttbk2^c.mut mice exhibited apparent locomotor deficiencies when observed in their cage (*Figure 1—video 1* ). To further examine motor coordination in Ttbk2^c.mut mice, we employed a rotarod performance test. Ttbk2^c.mut mice exhibited a shorter latency to fall compared to the littermate Controls in each trial, for both the accelerating rotarod analysis as well as the steady speed rotarod analysis (*Figure 1A,B*). These results indicate that Ttbk2^c.mut mice are impaired in their motor coordination, consistent with motor deficits observed in multiple mouse models of SCA (*Lalonde and Strazielle, 2019*; *Klockgether et al., 2019*).

To assess whether the motor behavioral changes we observed in the Ttbk2^c.mut animals are a consequence of changes to neuronal architecture in the adult brain, we examined Ttbk2^c.mut mice at 4 months of age (3 months post TMX). The brains of Ttbk2^c.mut mice have slightly smaller olfactory bulbs, but the overall gross morphology of the cortex and cerebellum was unchanged (*Figure 1—figure supplement 1H*). SCA11 pathology is associated with degeneration of the cerebellar neurons. We therefore examined the architecture and connectivity of neurons within the cerebellum to assess whether the Ttbk2^c.mut animals exhibited phenotypes similar to those described for mouse models of other subtypes of SCA. Within the cerebellum, PCs are the major source of functional neuronal output, and receive excitatory inputs primarily from parallel fibers and climbing fibers. Parallel fibers extend from the granule neurons, a population of densely packed neurons found directly beneath PCs (*Ichikawa et al., 2016*). Climbing fibers extend from neurons of the Inferior Olivary Nuclei (ION) in the medulla (*Kano et al., 2018*). These connections are essential for PC function, and dysfunction or loss of these connections, particularly the VGLUT2+ excitatory synapses from the climbing fibers, has been shown in various mouse models of SCA to be linked to pathology and disease progression (*Duvick et al., 2010*; *Ebner et al., 2013*; *Furrer et al., 2013*; *Smeets and Verbeek, 2016*; *Smeets et al., 2015*).

To assess the changes throughout Ttbk2^c.mut cerebella, we looked at the different layers of the cerebellar folia. We observed thinning of the molecular layer of the cerebellum, which comprises elaborate dendrites extended from the PCs (*Figure 1C,D*. 175 µm ± 3.422 for Control vs. 160.9 µm ± 2.527 for Ttbk2^c.mut). More dramatically, on examination of the synaptic marker VGLUT2, we found

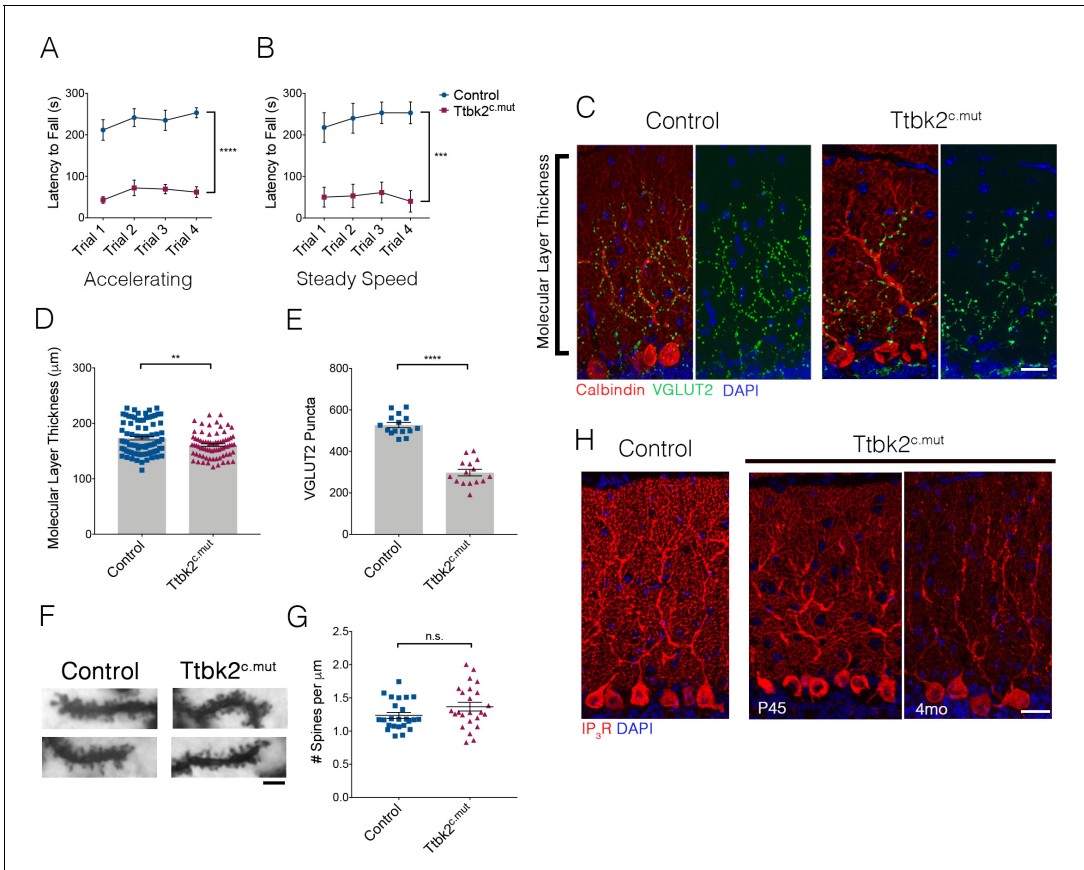

**Figure 1.** Loss of *Ttbk2* causes SCA-like phenotypes. (**A, B**) Accelerating and steady speed rotarod performance test between Ttbk2[c.mut] and littermate Controls. Ttbk2[c.mut] animals have a shorter latency to fall time in both tests, indicative of impaired motor ability (a two-way ANOVA with Bonferroni's multiple comparison test was used for calculating significance. p<0.0001 for accelerating rotarod test, and p=0.0001 for steady speed. n = 9 animals for Control, n = 8 animals for Ttbk2[c.mut]). (**C**) Cerebellar tissue from Control and Ttbk2[c.mut] mice at 3 months after loss of *Ttbk2*, immunostained for Calbindin to label Purkinje cells (red) and VGLUT2 to show climbing fiber synapses (green). Ttbk2[c.mut] animals show a reduction in VGLUT2 positive synapses throughout the cerebellum 3 months after loss of TTBK2. Scale bar = 50 µm. (**D**) Quantification of molecular layer length in Ttbk2[c.mut] cerebellar tissue (each point represents one measurement, 75 measurements overall. n = 3 animals. p=0.0011 by student's unpaired t-test, error bars indicate SEM). (**E**) Quantification of VGLUT2+ puncta throughout PC dendrites. Ttbk2[c.mut] animals show a significant reduction in these VGLUT2+ synapse terminals (each point represents one measurement, 15 measurements per genotype, n = 3 animals. p<0.0001 by student's unpaired t-test, error bars indicate SEM). (**F**) Golgi stain showing spines on proximal dendrites of PCs in Control and Ttbk2[c.mut] animals. Scale bar = 2 µm. (**G**) Quantification of number of spines per micron of dendrite. Ttbk2[c.mut] PCs do not lose spine density on proximal dendrites at 4 months of age. Each point represents a measurement taken from a singular dendrite, n = 3 animals. (**H**) Immunostaining for IP3R (red) and nuclei (blue). Loss of IP3R expression is seen as early as P45 in Ttbk2[c.mut] cerebellum. By 3 months after TMX injection, IP3R expression is no longer localized to secondary dendrites throughout the dendritic tree of PCs in Ttbk2[c.mut] animals. Scale bar = 50 µm.

The online version of this article includes the following video and figure supplement(s) for figure 1:

**Figure supplement 1.** Ttbk2[c.mut] animals have phenotypes shared with other ciliopathy models.

**Figure supplement 2.** Dendritic trees of Ttbk2[c.mut] PCs do not exhibit gross morphological changes.

**Figure supplement 3.** Loss of *Ttbk2* starting at P45 results in loss of VGLUT2 synapses.

**Figure 1—video 1.** Ttbk2[c.mut] mice have apparent motor coordination deficiencies.

https://elifesciences.org/articles/51166#fig1video1

a marked reduction in these puncta throughout the Ttbk2[c.mut] cerebellum compared to Controls (*Figure 1C,E* 527.3 puncta ± 12.68 for Control vs. 297.8 puncta ± 15.65 for Ttbk2[c.mut]) despite a lack of obvious gross morphological changes in the dendritic arbors of Ttbk2[c.mut] PCs (*Figure 1—figure supplement 2A,B*). Additionally, Ttbk2[c.mut] animals injected with tamoxifen beginning at P45 exhibit the same loss of VGLUT2+ synapses (*Figure 1—figure supplement 3A-B*), showing that loss of TTBK2 after all connections and development of the cerebellum are complete, leads to

degeneration of these synapses. We confirmed that this loss of VGLUT2 synapses was not caused by loss of dendritic spines, with use of a Golgi stain to view individual dendrites and spines on Ttbk2$^{c.mut}$ PCs (*Figure 1F,G*; 1.236 ± 0.22 spines per µm for Control vs. 1.368 ± 0.32 spines per µm for Ttbk2$^{c.mut}$).

To further explore the role of TTBK2 in PC function, we examined calcium receptor abundance in these neurons. PCs require an intracellular calcium modulation network as a means of signaling function. Within this network inositol 1,4,5-trisphosphate receptors (IP3Rs) are key calcium channel regulators needed for calcium release from the surrounding endoplasmic reticulum (ER) throughout the PC (*Sarkisov and Wang, 2008*). Precise regulation of IP3R activity is critical, and a balance of calcium channel release is imperative to the overall function of the cerebellum. Mutations in the IP3R1 gene have been linked to SCA15 and SCA29, while overexpression of IP3R1 underlies phenotypes within SCA2 and SCA3 (*Tada et al., 2016*). Because of these links to other SCA-related phenotypes, we therefore examined IP3R expression throughout the Ttbk2$^{c.mut}$ animals. We found that levels of IP3R in PCs are reduced in Ttbk2$^{c.mut}$ compared to Controls starting at P45, with expression strongly reduced throughout the PCs 3 months after TMX (*Figure 1H*). Thus, like mouse models of other SCA subtypes, the PCs of Ttbk2$^{c.mut}$ animals exhibit defects in calcium modulation consistent with dysfunction of these cells.

## Loss of *Ttbk2* causes changes to ION neurons and BG

Next, we tested whether loss of *Ttbk2* affects other cell types linked to the pathology of SCA, in addition to the PCs. Climbing fibers extend from neurons of the inferior olivary nucleus (ION) in the medulla. These fibers traverse the brain stem, enter the cerebellar cortex, and innervate the PC dendrites (*Watanabe and Kano, 2011*). As we saw a reduction of the VGLUT2+ synaptic terminals between these climbing fibers and PC dendrites, we examined the soma of the ION neurons from which these climbing fibers extend. In several subtypes of SCA, including SCA1, 2, 3, 6, and 7 (*Seidel et al., 2012*), the pathology of the disorder is characterized in part by the loss of ION soma; a characteristic also observed in mouse models of these diseases. Neurons within the ION can be identified by dual expression of Calbindin and NeuN in the medial ventral region of the medulla (*Figure 2—figure supplement 1A,B*). When we looked at this population of neurons, we did not notice a loss of these cells. However, we did find that the perikarya of the neurons within the ION were smaller in Ttbk2$^{c.mut}$ animals compared to Controls. We therefore used NeuN to label the perikarya of neurons within the ION and measured the area of the somata of ION neurons, and found a significant reduction in the area of ION neuron soma in Ttbk2$^{c.mut}$ mice at 4 months of age compared to Controls (*Figure 2A,B*; 180.4 µm$^2$ ± 1.93 for Control vs. 109 µm$^2$ ± 1.01 for Ttbk2$^{c.mut}$). Neuronal shrinkage is a phenotype that has been noted in patients with SCA1 as well as Friedrich's ataxia, and is thought to precede neuronal apoptosis (*Nagaoka, 2003*; *Dell'Orco et al., 2015*; *Kemp et al., 2016*). This implies that, in addition to the PCs themselves, the neurons sending critical inputs to the PCs are perturbed in Ttbk2$^{c.mut}$ mice.

Throughout the brain, astrocytes and glia also play important roles in maintaining synaptic connectivity and strength. In the cerebellum, the processes of the BG are interspersed with PC dendrites in the molecular layer, with BGs enwrapping the excitatory synapses of the PCs (*Leung and Li, 2018*). As defects in BG morphology have been linked to the etiology of SCA7 (*Furrer et al., 2011*), we examined the BGs in Ttbk2$^{c.mut}$. To assess the morphology of BGs in the Ttbk2$^{c.mut}$ animals and evaluate whether defects in these cells may contribute to the phenotype, we used GFAP to visualize BG fibers that extend throughout the cerebellar folia. We found that the numbers of glial fibers were moderately reduced in Ttbk2$^{cmut}$ cerebellar folia compared to littermate Controls (*Figure 2C, D*; 11.44 BG fibers ± 0.29 for Control vs. 7.64 BG fibers ± 0.22 for Ttbk2$^{cmut}$), suggesting that loss of *Ttbk2* has modest effects on the morphology of BGs. Taken together, these data suggest that loss of *Ttbk2* affects several cell types in the cerebellum and medulla, underscoring the widespread importance of *Ttbk2* within these tissues.

## TTBK2 is required cell-autonomously in PCs to maintain their connectivity

Dysfunction and eventual atrophy of the PCs in the cerebellum is the primary pathology underlying SCA11 in human patients (*Houlden et al., 2007*). In our conditional *Ttbk2* mutant mice, the most

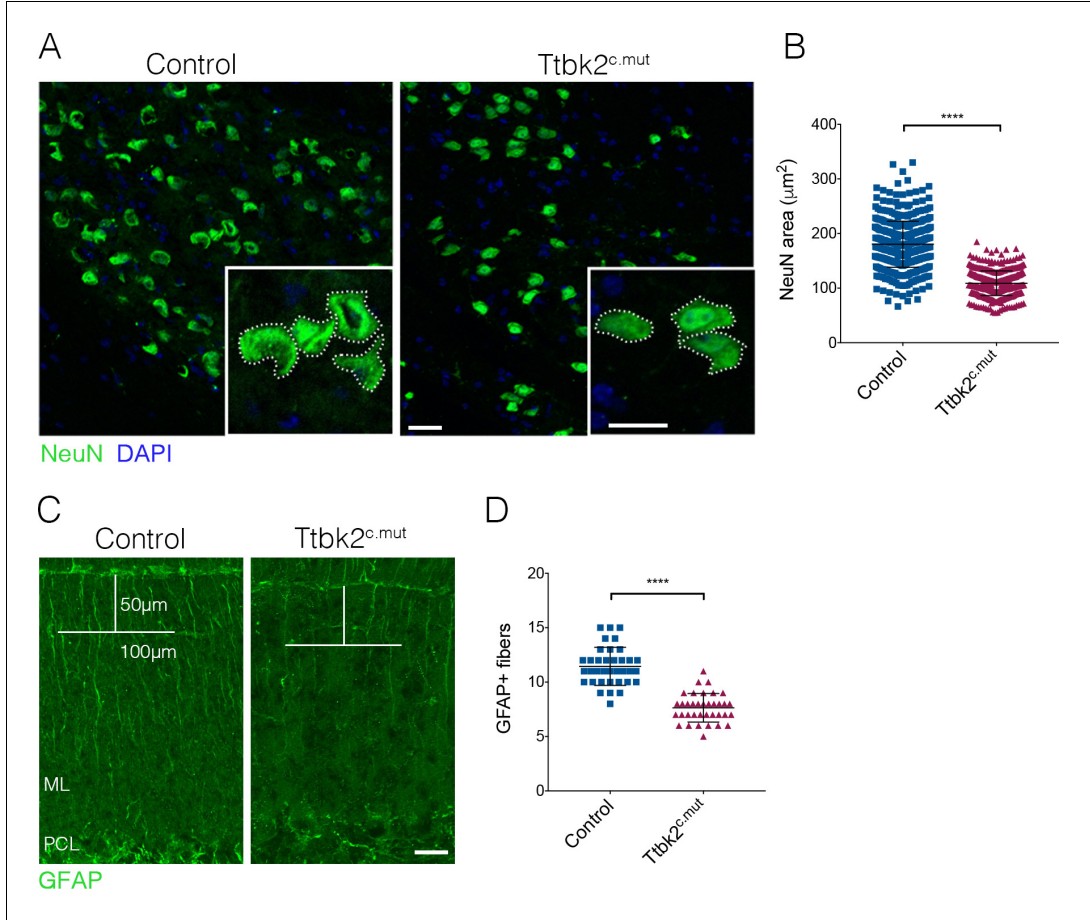

**Figure 2.** ION and glial cells are affected by loss of *Ttbk2*. (**A**) Representative images of neurons in the inferior olivary nucleus (ION) located in the medulla. Neural somata are immunostained with NeuN (green). Insets show how the area was measured. Scale bar = 50 µm (20 µm inset). (**B**) Quantification of NeuN area. ION neurons have reduced area in Ttbk2$^{c.mut}$ animals compared to Control (each point represents a single cell measurement of which > 150 measurements were made per animal. n = 3 animals, p<0.0001 by unpaired student's t-test, error bars indicate SEM). (**C**) Glial fibrillary acidic protein (GFAP) staining showing BG fibers throughout the molecular layer. In Ttbk2$^{c.mut}$ animals, density of these fibers is reduced. Quantification was made as previously described (*Furrer et al., 2011*), in which a 50 µm line was drawn from the pial surface of the folia, and a 100 µm line across. Glial fibers that fully crossed the 100 µm line were scored. Scale bar = 20 µm. (**D**) Quantification of GFAP+ glial fibers that crossed the 100 µm line (each point represents an image quantified, 36 images quantified per genotype across n = 3 animals. p<0.0001 by unpaired student's t-test, error bars indicate SEM).

The online version of this article includes the following figure supplement(s) for figure 2:

**Figure supplement 1.** Identification of ION neurons within the medulla.

prominent phenotype is altered connectivity of the PCs with additional cellular changes seen in the BGs as well as ION neurons. To determine the degree to which these defects are the result of cell autonomous vs. non-cell-autonomous requirements for *Ttbk2* in the PCs, we used the PC-specific Cre line *Pcp2-Cre*, which drives recombination specifically in PCs within the cerebellum beginning at P6 (*Zhang et al., 2004*). At P30, *Ttbk2$^{fl/fl}$;Pcp2-Cre$^{+}$* (referred to from here as Ttbk2$^{Pcp2}$) animals have normal cerebellar structure throughout, with molecular layer thickness comparable to that of littermate Control animals (*Figure 3A,C*. 202.7 µm ± 3.51 in P30 Control vs. 191.5 µm ± 3.21 in P30 Ttbk2$^{Pcp2}$). The VGLUT2+ synapses between climbing fibers and PCs are not significantly changed between P30 Control and Ttbk2$^{Pcp2}$ animals (*Figure 3A,D*; 548.2 puncta ± 13.36 in P30 Control vs. 538.9 puncta ± 18.14 in P30 Ttbk2$^{Pcp2}$). This indicates that despite postnatal loss of *Ttbk2*, initial connections between PCs and climbing fibers are established normally. By P90, however, Ttbk2$^{Pcp2}$ animals exhibited phenotypes largely recapitulating those observed in the Ttbk2$^{c.mut}$ animals. At P90,

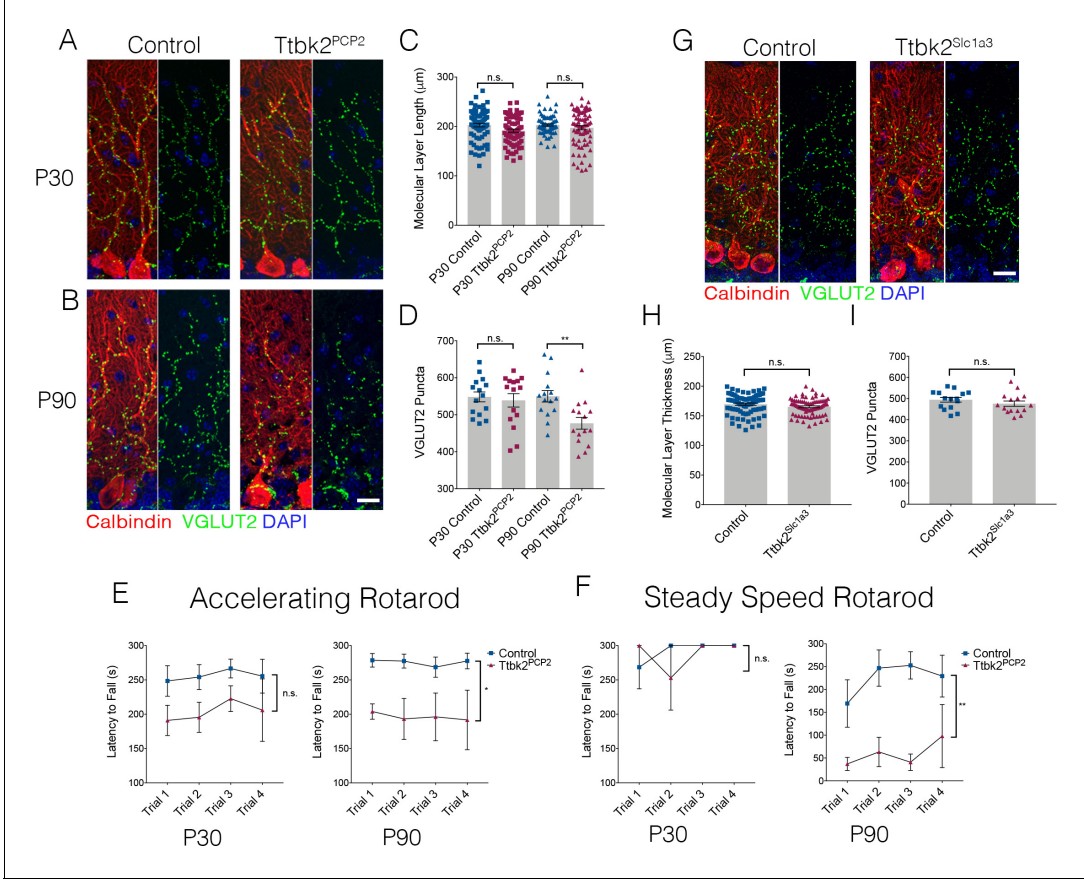

**Figure 3.** Cell autonomous requirements for *Ttbk2* in the cerebellum. (**A,B**) Representative images of Control and *Ttbk2^f/f^;Pcp2Cre+* (Ttbk2^Pcp2^) animals at age P30 (**A**) and P90 (**B**), immunostained for Calbindin to label PCs (red), VGLUT2 to label synapses (green), and nuclei (blue). VGLUT2 terminals are reduced in P90 Ttbk2^Pcp2^ animals compared to P30 Ttbk2^Pcp2^ animals. Scale bar = 20 μm. (**C**) Quantification of molecular layer thickness in P30 and P90 Ttbk2^Pcp2^ and Control animals (each point represents one measurement, 75 measurements per genotype. n = 3 animals. No significant difference reported by one-way ANOVA with Tukey correction, error bars indicate SEM. (**D**) Quantification of VGLUT2+ puncta analysis in P30 and P90 Ttbk2^Pcp2^ and Control animals. There are no differences in the number of puncta at P30; however these are significantly reduced by P90 (each point represents one field analyzed, five fields analyzed per animal, n = 3 animals. p=0.0098 by one-way ANOVA with Tukey correction, error bars indicate SEM). (**E**) Accelerating rotarod performance test of Ttbk2^Pcp2^ and littermate Controls from P30 to P90. Ttbk2^Pcp2^ animals have a significantly shorter latency to fall time at P90 compared to P30 (a two-way ANOVA with Bonferroni's multiple comparison test was used for calculating significance. p=0.1051 for P30 accelerating rotarod test, and p=0.0161 for P90 accelerating rotarod test). (**F**) Steady speed rotarod performance test of Ttbk2^Pcp2^ and littermate Controls aging from P30 to P90. At P30 Ttbk2^Pcp2^ animals do not have a shorter latency to fall time compared to Controls on the steady speed rotarod. However, by P90 there is a drastic reduction in latency to fall time for Ttbk2^Pcp2^ animals compared to Controls, indicative of impaired motor ability with age (a two-way ANOVA with Bonferroni's multiple comparison test was used for calculating significance. p=0.7819 for P30 steady speed rotarod test, and p=0.0023 for P90 steady speed rotarod test. n = 6 animals for Control, n = 4 animals for Ttbk2^Pcp2^). (**G**) Representative images of Control and *Ttbk2fl/fl;Slc1a3-CreER* (Ttbk2^Slc1a3^) animals at 4 months of age (3 months post TMX) treatment, immunostained for Calbindin to label PCs (red), VGLUT2 to label synapses (green) and nuclei (blue). Unlike Ttbk2^c.mut^ and Ttbk2^Pcp2^ mice, there is no loss of VGLUT2 synapses throughout the PC dendrites of Ttbk2^Slc1a3^ mice relative to Controls. (**H**) Quantification of molecular layer length in Ttbk2^Slc1a3^ and Control animals (each point represents one measurement, 75 measurements per genotype. n = 3 animals. No significance reported by student's unpaired t-test, error bars indicate SEM). (**I**) Quantification of VGLUT2+ puncta analysis in Ttbk2^Slc1a3^ and Control animals. There is no difference in the numbers of puncta between these conditions (each point represents a field analyzed, five images analyzed per animal, n = 3 animals. No significance reported by student's unpaired t-test, error bars indicate SEM).

The online version of this article includes the following figure supplement(s) for figure 3:

**Figure supplement 1.** Cilia are lost from PCs in Ttbk2^Pcp2^ cerebellum.

numbers of primary cilia are significantly reduced on PCs of Ttbk2$^{Pcp2}$ animals (*Figure 3—figure supplement 1A,B*; 48.93 ± 7.86 percent PCs ciliated in Control vs. 18.67 ± 11.07 percent PCs ciliated in P90 Ttbk2$^{Pcp2}$). While molecular layer thickness between the P90 Control and Ttbk2$^{Pcp2}$ was not changed (*Figure 3B,C*; 202.9 μm ± 2.11 in P90 Control vs. 197.4 μm ± 4.18 in P90 Ttbk2$^{Pcp2}$), we see a significant decrease in VGLUT2 puncta throughout the cerebellum (*Figure 3B,D*; 549.9 puncta ± 15.47 in P90 Control vs. 476.9 puncta ± 15.82 in P90 Ttbk2$^{Pcp2}$), indicating that PCs have started to lose these important connections from the climbing fiber synapses.

We then assessed motor coordination of P30 and P90 Ttbk2$^{Pcp2}$ animals using the rotarod performance test and did not observe significant changes in P30 animals. However, by P90, the Ttbk2$^{Pcp2}$ animals consistently exhibited reduced latency to fall on both the accelerating rotarod as well as the steady speed rotarod performance tests (*Figure 3E and F*). These data show that loss of *Ttbk2,* specifically from PCs, causes neurodegenerative phenotypes.

Our data from the *Ttbk2* global conditional knockouts revealed that the morphology of BGs was modestly perturbed (*Figure 2C,D*). As defects in BGs have been shown to non-cell autonomously contribute to the degenerative phenotypes observed in SCA7 (*Furrer et al., 2011*), we tested whether deletion of TTBK2 specifically from these cells could also result in loss of synapses and other degenerative changes to the PCs. We crossed *Ttbk2$^{fl/fl}$* animals to a *Slc1a3-CreER* mouse (*Wang et al., 2012*) to produce *Ttbk2$^{fl/fl}$; Slc1a3-CreER+* mice to induce recombination of the *Ttbk2* allele specifically within glial cells. Following the same TMX injection protocol used for the Ttbk2$^{c.mut}$ experiments, we did not see changes to the VGLUT2+ synapses on PC dendrites (*Figure 3G–I*). These data indicate that the PC phenotypes observed in the Ttbk2$^{c.mut}$ mice are primarily cell autonomous.

## Conditional knockout of *Ttbk2* recapitulates SCA11 phenotypes

In the first 3 months following TMX injections, the phenotypes exhibited by the Ttbk2$^{c.mut}$ mice consisted mainly of altered synaptic connectivity between PC and ION climbing fibers, and accompanying deficits in motor coordination (*Figure 1*). However, when we assessed the cerebellar phenotypes of animals at 6 months of age (5 months following TMX injection), we found gaps in the molecular layer where PCs appear to be absent (*Figure 4A*). We quantified this observation by counting PC soma within a defined region of the primary fissure, and confirmed that the number of PCs is reduced in 6-month-old Ttbk2$^{c.mut}$ mice compared to littermate Controls of the same age, as well as compared to 4-month-old Ttbk2$^{c.mut}$ mice (*Figure 4B*; 18.5 ± 0.29 PCs per 500 μm for 4-month Control vs. 18.42 PCs ± 0.34 for 4-month Ttbk2$^{c.mut}$; 18.67 PCs ± 0.43 for 6-month Control vs. 11.92 PCs ± 0.74 for 6-month Ttbk2$^{c.mut}$). We found more PC gaps in folia of 6-month-old Ttbk2$^{c.mut}$ animals compared to Controls, and that most gaps are enriched at the inner folia. PC gaps were not found on folia X, which is consistent with data showing folia X being resistant to neurodegeneration (*Figure 4C* and *Figure 4—figure supplement 1A*) (*Tolbert et al., 1995*).

A postmortem examination of a SCA11 affected individual revealed Tau aggregates in regions of the brain outside of the cerebellum (*Houlden et al., 2007*). We therefore looked for pathological Tau aggregates in the cortex in both the 4-month-old and 6-month-old Ttbk2$^{c.mut}$ animals. We could not detect the accumulation of phosphorylated Tau in the Control or 4-month-old Ttbk2$^{c.mut}$ cortex. However, in the cortex of the 6-month-old Ttbk2$^{c.mut}$ animals, we noticed a small number of neurons with some accumulation of phosphorylated Tau, recognized by an antibody specific to Ser202 and Thr205 phosphorylated tau (*Figure 4—figure supplement 2A*). This mild accumulation is compared to neurons in a mouse model of Alzheimer's, JNPL3(P301L), which expresses a mutated form of Tau (*Lewis et al., 2000*) and is therefore positive for accumulation of phosphorylated Tau (*Figure 4—figure supplement 2B*). Thus, dysfunction of PCs after loss of *Ttbk2,* accompanied by Tau accumulation outside of the cerebellum, recapitulates currently described SCA11 phenotypes.

## Ttbk2$^{c.mut}$ animals lose neuronal primary cilia prior to the onset of neurodegenerative phenotypes

Our prior work demonstrated that mutations associated with SCA11, which result in the production of a truncated protein, interfere with the function of full-length TTBK2. In particular, these mutations dominantly interfere with cilia formation in embryos and cultured cells (*Bowie et al., 2018*). Throughout the adult cerebellum and other regions of the hindbrain, neurons possess primary cilia (*Figure 5A–C*). Within 20 days following administration of TMX to induce recombination (P45), the number of ciliated

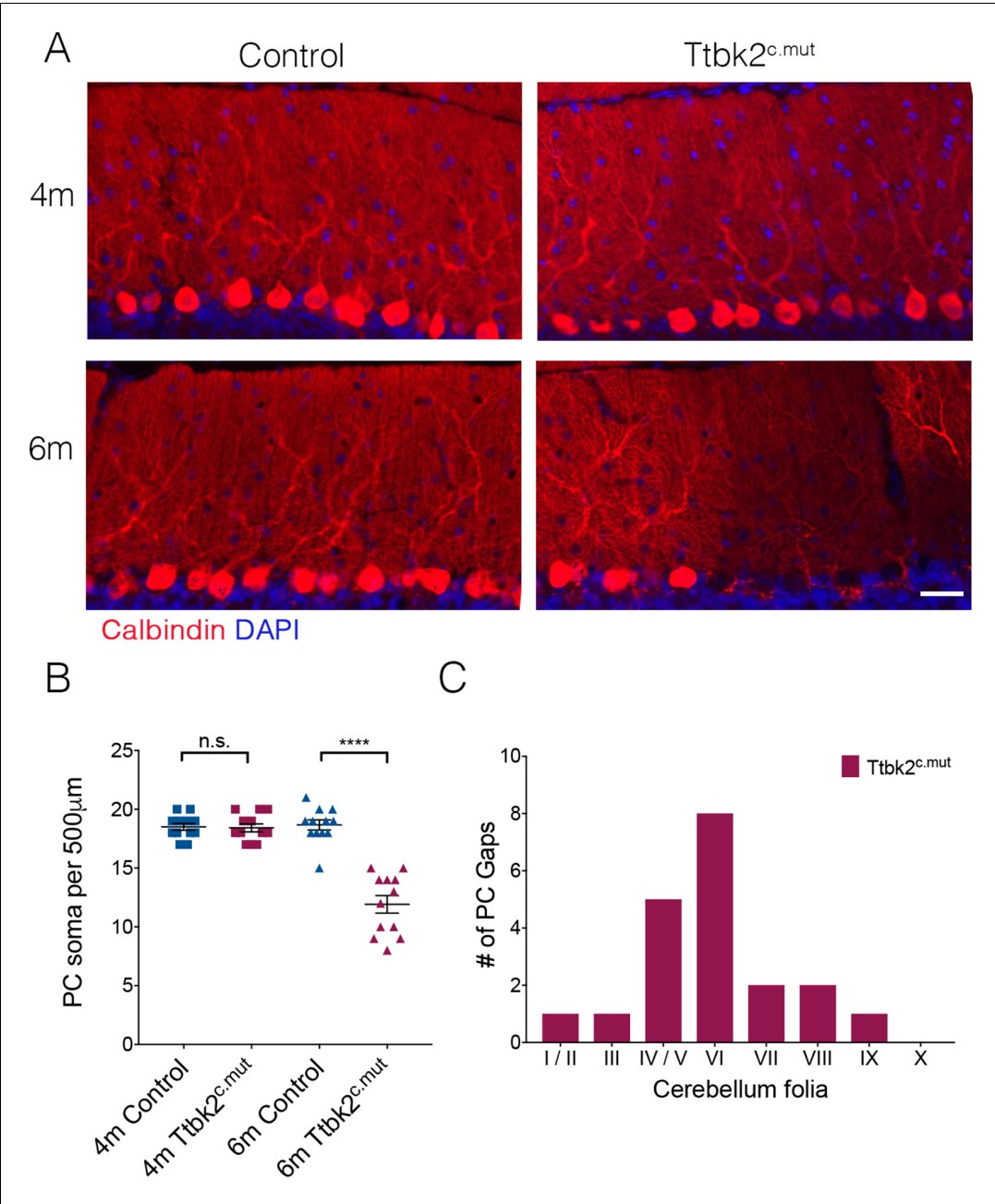

**Figure 4.** Aged Ttbk2[c.mut] animals lose Purkinje cells. (**A**) Representative images showing folia of 4-month-old Ttbk2[c.mut] (top) and 6-month-old Ttbk2[c.mut] animals (bottom) with respective littermate Controls. Cerebellum tissue is stained for Calbindin to show PC. 6-month-old Ttbk2[c.mut] have large stretches of folia missing Calbindin+ PC soma compared to 4-month-old Ttbk2[c.mut]. Scale bar 50 µm. (**B**) Quantification of the loss of PC soma along 500 µm stretch of the primary fissure (n = 36 measurements across three animals. p<0.0001 by student's unpaired t-test, error bars indicate SEM). (**C**) Quantification of location of PC gaps in Ttbk2[c.mut] animals. PC gaps are present throughout folia I-IX in Ttbk2[c.mut] animals but are not seen in Controls. A threshhold of more than two cell spaces was used to define a PC gap. 10 cerebellar slices were quantified, n = 3 animals.

The online version of this article includes the following figure supplement(s) for figure 4:

**Figure supplement 1.** PC gaps are present throughout most cerebellar folia in 6-month -old Ttbk2[c.mut] animals.
**Figure supplement 2.** Neurons in the cortex of 6-month-old Ttbk2[c.mut] animals have some Tau accumulation.

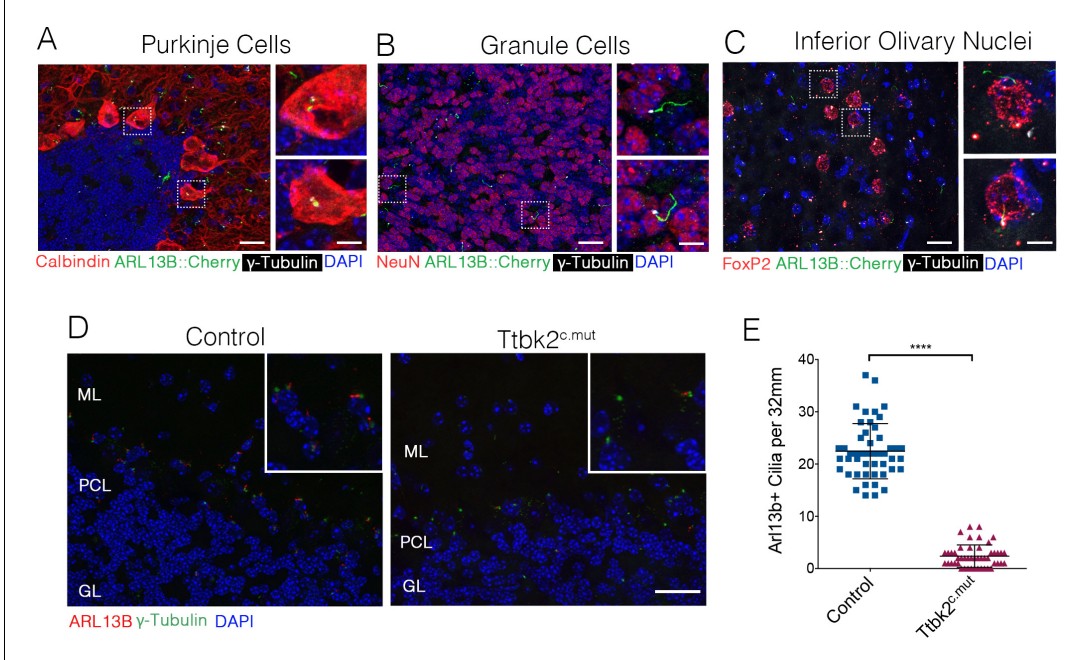

**Figure 5.** *Ttbk2* is critical for primary cilia stability on neurons. (A–C) Representative images of cilia on indicated cell types throughout parts of the cerebellum and medulla. Sections from a mouse expressing an ARL13B-mCherry transgene (green) (***Bangs et al., 2015***) immunostained for γ-Tubulin to label centrosomes (white), and various cell specific markers such as Calbindin to label Purkinje cells (A), NeuN to label granule neurons (B), and FoxP2 to label neurons within the inferior olivary nucleus (C). Insets show boxed areas. Scale bar = 50 µm, 10 µm for insets. (D) Representative images illustrating cilia loss in the cerebellum 20 days after TMX treatment. Sections were immunostained for ARL13B to label cilia (red) and γ-Tubulin to label centrosomes (green). For quantification purposes images were taken at the nexus between the molecular layer (ML) and granule layer (GL) with the r PCL in the middle of the imaging field where there is an abundance of cilia. Scale bar = 50 µm. (E) Quantification of cilia loss after TMX treatment (n = 36 images counted, three animals, p<0.0001 student's unpaired t-test, error bars indicate SEM).

The online version of this article includes the following figure supplement(s) for figure 5:

**Figure supplement 1.** Cilia are lost throughout the brain of Ttbk2$^{c.mut}$ animals.

cells in the cerebellum declined dramatically in Ttbk2$^{c.mut}$ mice: from a mean of 22.46 cilia per 32 mm$^2$ field ± 0.7626 in Control animals to 2.36 cilia per 32 mm$^2$ field ± 0.3103 in Ttbk2$^{c.mut}$ animals (***Figure 5D,E***). This loss of cilia was observed throughout the cerebellum, brain stem, and other areas of the brain such as the hippocampus and the cortex (***Figure 5—figure supplement 1A,B***). Thus, loss of cilia coincides with the behavioral changes we identified in Ttbk2$^{c.mut}$ mice, yet precedes the cellular changes occurring throughout the cerebellum of Ttbk2$^{c.mut}$ mice as they age.

## Loss of the cilium assembly gene *Ift88* recapitulates Ttbk2$^{c.mut}$ phenotypes

*Ttbk2* is essential both for the initiation of cilium assembly as well as the structure and stability of cilia (***Bowie et al., 2018***; ***Goetz et al., 2012***). Given this critical link between TTBK2 and primary cilia in all cell types examined in both developing and adult tissues, we tested whether loss or dysfunction of cilia via a different genetic mechanism causes convergent phenotypes to those of the Ttbk2$^{c.mut}$ mice. For these studies we turned to conditional mutants of another key ciliary protein, Intraflagellar Transport Protein 88 (IFT88). IFT88 is a component of the IFTB particle required for assembly of the ciliary axoneme as well as anterograde trafficking within the cilium (***Pazour et al., 2000***). Our previous work shows that IFT88 functions downstream of TTBK2 in cilium initiation (***Goetz et al., 2012***), with TTBK2 being required for IFT recruitment. In the developing and postnatal brain, IFT88 is important for cilia structure in the hippocampus and cortex (***Willaredt et al., 2008***) and when knocked out in these specific neuron populations results in memory deficits (***Berbari et al., 2014***). Additionally, *Ift88* null mutants exhibit nearly identical embryonic phenotypes to those of *Ttbk2* null mutants (***Murcia et al., 2000***). When we knocked out *Ift88* using the same approach described for

Ttbk2$^{c.mut}$ animals, we observed that the numbers of cilia were significantly reduced in Ift88$^{c.mut}$ cerebella at 3 months post TMX treatment, although more cilia remain in Ift88$^{c.mut}$ cerebella compared to the Ttbk2$^{c.mut}$ animals with the same treatment (*Figure 6A,C*; 17.31 cilia per 32 mm$^2$ field ± 0.65 for Control vs. 12.13 cilia per 32 mm$^2$ field ± 0.50 for Ift88$^{c.mut}$). Western blot analysis of cerebellar tissue from Ift88$^{c.mut}$ mice reveals that a small amount of IFT88 protein perdures in brain tissue (*Figure 6B*). This could, in part, help to explain why we do not see a full loss of cilia throughout the cerebellum similar to that observed in the Ttbk2$^{c.mut}$ mice. Regardless, the cilia that do remain in Ift88$^{c.mut}$ animals are shorter in length than Controls (*Figure 6D*; 2.31 µm ± 0.10 for Control vs. 1.70 µm ± 0.08 for Ift88$^{c.mut}$).

To further characterize the remaining cilia in the brains of Ift88$^{c.mut}$ mice, we examined additional markers of the ciliary membrane, including adenylate cyclase 3 (AC3) (*Guadiana et al., 2016*). We observed that within the WT cerebellum, there exist cilia that are AC3+ as well as AC3+/ARL13B+ (*Figure 6E*, arrowhead, *Figure 6F*). In our Ift88$^{c.mut}$ animals, we observed that the numbers of AC3+ cilia were strongly reduced (*Figure 6G*: 7.47 AC3+ cilia per 32 mm$^2$ field ± 0.41 in Control vs. 2.17 AC3+ cilia per 32 mm$^2$ field ± 0.22 in Ift88$^{c.mut}$). This analysis suggests that IFT88 is required for localization of specific signaling molecules such as AC3 to neuronal primary cilia throughout the cerebellum.

We then examined cerebellar structure and circuitry in Ift88$^{c.mut}$ animals. Similar to our findings in Ttbk2$^{c.mut}$ animals, no changes to PC number were evident at 3 months post TMX treatment. Molecular layer thickness was reduced in Ift88$^{c.mut}$ animals (*Figure 7A,B*; 182.3 µm ± 2.5 in Control vs. 169.3 µm ± 3.04 in Ift88$^{c.mut}$). Similarly, VGLUT2 puncta were reduced in Ift88$^{c.mut}$ compared to Controls (*Figure 7A,C*; 515.7 puncta ± 20.58 in Control vs. 395.6 puncta ± 13.7 in Ift88$^{c.mut}$). We also tested Ift88$^{c.mut}$ animals on the rotarod performance test to uncover any motor coordination deficits, given that these mice exhibit similar cellular changes to those of the Ttbk2$^{c.mut}$ animals. These tests revealed that Ift88$^{c.mut}$ animals also have a shorter latency to fall time on the steady speed rotarod performance test, but not on the accelerating rotarod performance test (*Figure 7D,E*), while Ttbk2$^{c.mut}$ animals have a shorter latency to fall time on both accelerating and steady speed rotarod performance tests (*Figure 1A,B*). Steady speed rotarod analysis is thought to more accurately detect motor coordination deficits, while the accelerating rotarod test can also be affected by mouse fatigue (*Monville et al., 2006*). Taken together, these data show that loss of IFT88 from the adult brain results in impaired ciliary structure and similar defects in cerebellar architecture and locomotor behavior to those observed in the animals lacking TTBK2.

We further assessed whether the Ift88$^{c.mut}$ animals also lose PCs as they age, as was the case for the Ttbk2$^{c.mut}$ mice (*Figure 4*). 6-month-old Ift88$^{c.mut}$ animals show gaps throughout the PC layer, and have reduced numbers of PC soma (*Figure 8A,B*; 17.5 ± 0.44 per 500 µm in 4-month-old Control vs. 16.92 PC soma ± 0.31 in 4-month-old Ift88$^{c.mut}$. 17.17 PC soma ± 0.55 in 6-month-old Control vs. 12.67 PC soma ± 0.43 in 6-month-old Ift88$^{c.mut}$). Coupled with these findings, the molecular layer thickness is further reduced in 6-month-old Ift88$^{c.mut}$ animals (*Figure 8C–E*; 173.9 µm ± 2.28 in 6-month-old Control vs. 158.0 µm ± 1.63 in 6-month-old Ift88$^{c.mut}$), as well as VGLUT2 puncta counts being diminished in 6-month-old Ift88$^{c.mut}$ (*Figure 8C,D,F*; 645.9 puncta ± 26.83 in 6-month-old Control vs. 461.4 puncta ± 25.42 in 6-month-old Ift88$^{c.mut}$).

## Discussion

In this work, we tested the hypothesis that SCA11 pathology results from the requirements for TTBK2 in cilium assembly and stability. We showed that TTBK2 is essential for maintaining the connectivity and viability of PCs in the adult cerebellum. These phenotypes are similar to those reported for mouse models of other subtypes of SCA as well as consistent with many aspects of SCA11 in human patients, suggesting that the Ttbk2$^{c.mut}$ mice model the human condition. We further demonstrated that mice conditionally lacking the ciliary protein IFT88 in adult tissues exhibit highly similar neurodegenerative phenotypes to those observed in the Ttbk2$^{c.mut}$ mice, including loss of excitatory synapses from the climbing fibers and the eventual loss of PCs. The high degree of convergence of these phenotypes suggests that the neural degenerative phenotypes of the Ttbk2$^{c.mut}$ mice are driven primarily by the requirement for TTBK2 in mediating cilium assembly, and points to a critical role for these organelles in maintaining neuronal function during adulthood.

Cilia and ciliary signals play a variety of important roles during embryonic and postnatal development of the brain and central nervous system. Cilia are linked to processes including the expansion

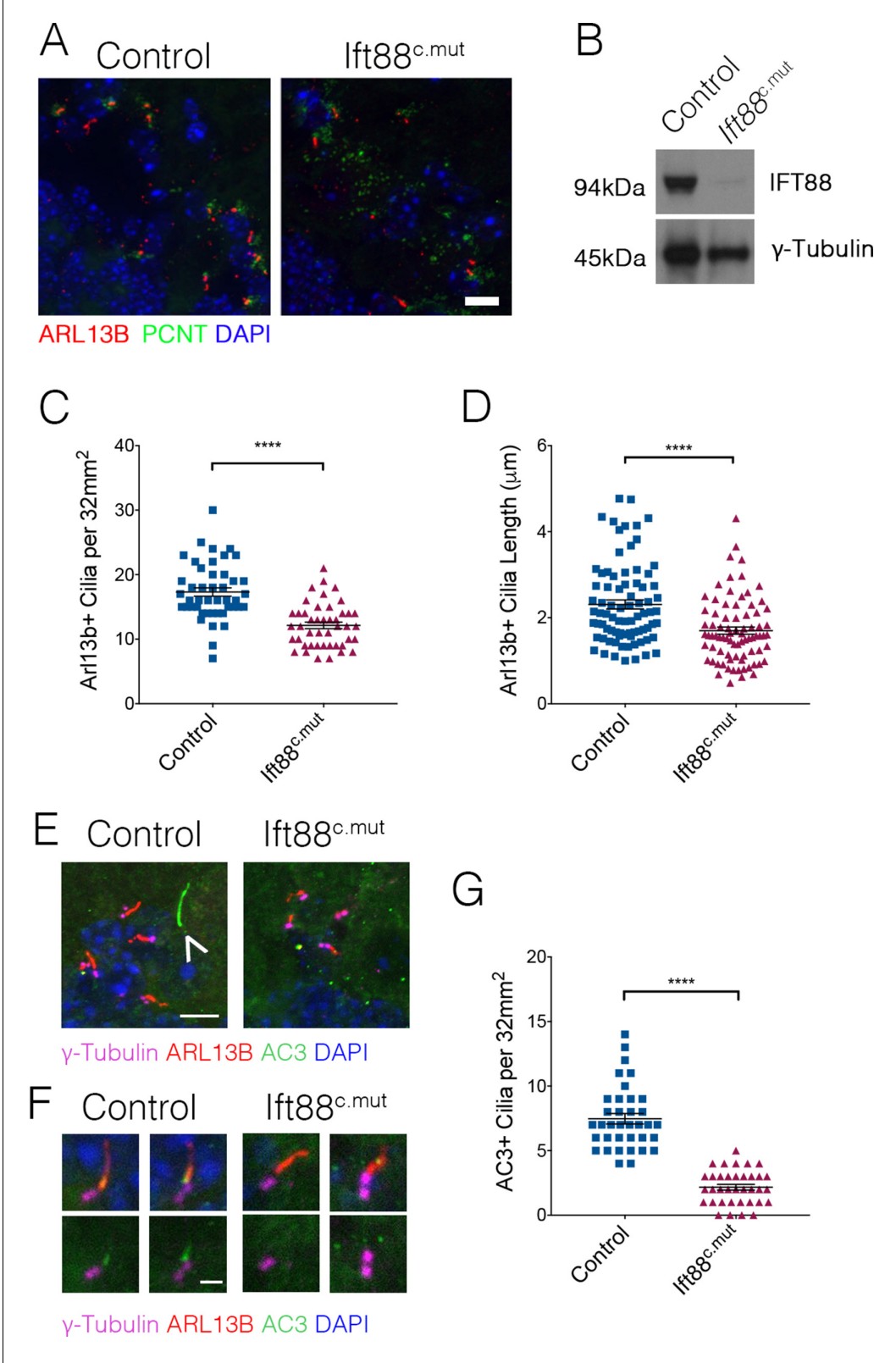

**Figure 6.** Ift88[c.mut] have fewer, shorter cilia throughout the cerebellum and mislocalization of ciliary membrane markers. (**A**) Representative images illustrating cilia loss in the cerebellum of Ift88[c.mut] animals, immunostained for ARL13B to label cilia (red), Pericentrin (PCNT) to label centrosomes (green) and nuclei (blue) Scale bar = 20 μm. (**B**) Western blot analysis of IFT88 in cerebellum lysate 3 months after TMX injections in Ift88[c.mut] animals. (**C**) *Figure 6 continued on next page*

*Figure 6 continued*

Quantification of cilia loss. Compared to Ttbk2[c.mut] mice, cilia loss is less dramatic in the cerebellum of Ift88[c.mut] animals (each point represents a field scored, 45 fields scored per genotype. n = 3 animals. p<0.0001 by student's unpaired t-test, error bars indicate SEM). (D) Quantification of cilia length between Control and Ift88[c.mut]. Cilia in Ift88[c.mut] cerebellum are shorter (each point represents a single cilium, 80 cilia were measured for each genotype. n = 3 animals, p<0.0001 by student's unpaired t-test, error bars indicate SEM). (E,F) Cilia from 6-month-old Control and Ift88[c.mut] stained for γ-Tubulin to label centrosomes (magenta), ARL13B to label cilia membrane (red), AC3 to label cilia membrane (green) and DAPI (blue). Ift88[c.mut] lose AC3+ cilia. (E) The arrow indicates a cilium that is AC3 +/ARL13B-. Scale bar = 5 µm (E) and 1 µm (F). (G) Quantification of AC3+ cilia throughout the cerebellum. Ift88[c.mut] animals have a strong reduction in AC3+ cilia localization (each point represents a field scored, 36 field scored per genotype. n = 3 animals. p<0.0001 by student's unpaired t-test, error bars indicate SEM).

and patterning of neural progenitors (*Guemez-Gamboa et al., 2014*), the migration and laminar placement of interneurons (*Higginbotham et al., 2012*), and in the establishment of neuronal morphology (*Guadiana et al., 2016*; *Sarkisian and Guadiana, 2015*). Consistent with the varied roles of cilia in neural development, an array of neurological deficits are among the most common hallmarks of ciliopathies (*Lee and Gleeson, 2010*; *Youn and Han, 2018*), highlighting the importance of these organelles in human health. In addition to their critical developmental functions, mounting evidence supports an important role for ciliary signaling in tissue regeneration and homeostasis in adult organs, including the kidneys (*Davenport et al., 2007*), skin (*Croyle et al., 2011*), skeletal muscle (*Kopinke et al., 2017*), and bone (*Moore et al., 2018*).

Within the adult CNS, dysfunction of ciliary trafficking is linked to retinal degeneration (*Wheway et al., 2014*). Degeneration of photoreceptors is a feature of many human ciliopathies as well as mouse models of these disorders (*Braun and Hildebrandt, 2017*; *Bujakowska et al., 2017*), and occurs as trafficking within the photoreceptor outer segments (the modified cilium) fails. This results in accumulation of rhodopsin within the cell body and leads to death of the photoreceptor neurons through mechanisms that are not completely understood (*Seo and Datta, 2017*). Within the brain, conditional loss of the ciliary protein ARL13B from mouse striatal interneurons, both during their development and following maturation, results in changes in their morphology and connectivity (*Guo et al., 2017*). Our work extends these findings and provides strong genetic evidence that primary cilia and signals mediated by these organelles are important to maintain the morphology and function of a specific type of neuron within the brain, cerebellar PCs. Of substantial interest for future studies in our laboratory is the question of whether additional types of neurons in other regions of the brain require primary cilia to maintain their connectivity or viability. These include neurons that are affected by other more common neurodegenerative conditions, such as hippocampal neurons affected by Alzheimer's disease, midbrain dopaminergic neurons lost in Parkinson's disease, or medium spiny neurons affected in Huntington's disease. In each of these cases, links have been made between cilia dysfunction and these disorders, although functional and mechanistic studies have yet to be performed (*Chakravarthy et al., 2012*; *Dhekne et al., 2018*; *Keryer et al., 2011*).

Our data show that conditional mutants of *Ttbk2* and *Ift88* have similar phenotypes with respect to loss of excitatory synapses to PCs from the climbing fibers, motor coordination deficits, and eventually loss of PCs. This evidence suggests that these defects are the result of ciliary loss or dysfunction. We note, however, that the ciliary phenotypes that result from loss of *Ttbk2* differ from those observed in the *Ift88* conditional mutants in the context of the adult brain. *Ttbk2* conditional mutant mice rapidly lose cilia following administration with TMX, with nearly all cilia within the cerebellum being absent within 20 days. In contrast, the numbers of cilia in the *Ift88* conditional mutants are only slightly reduced 3 months following TMX. However, these cilia exhibit significant abnormalities, including reduced length, and a near-complete loss of AC3 from the remaining cilia. This suggests that the degenerative phenotypes observed in both conditional mutants are driven by the loss of a specific ciliary signal.

While the precise nature of the ciliary signals that maintain the connectivity and viability of PC neurons remains unknown, there are a number of candidates. Many different GPCRs and associated signaling cascades and second messengers have been shown to concentrate in primary cilia or to be enriched predominantly at the primary cilium (*Mykytyn and Askwith, 2017*). In particular, cAMP and Ca++ are highly concentrated within the cilium (*Moore et al., 2016*). Misregulation of these

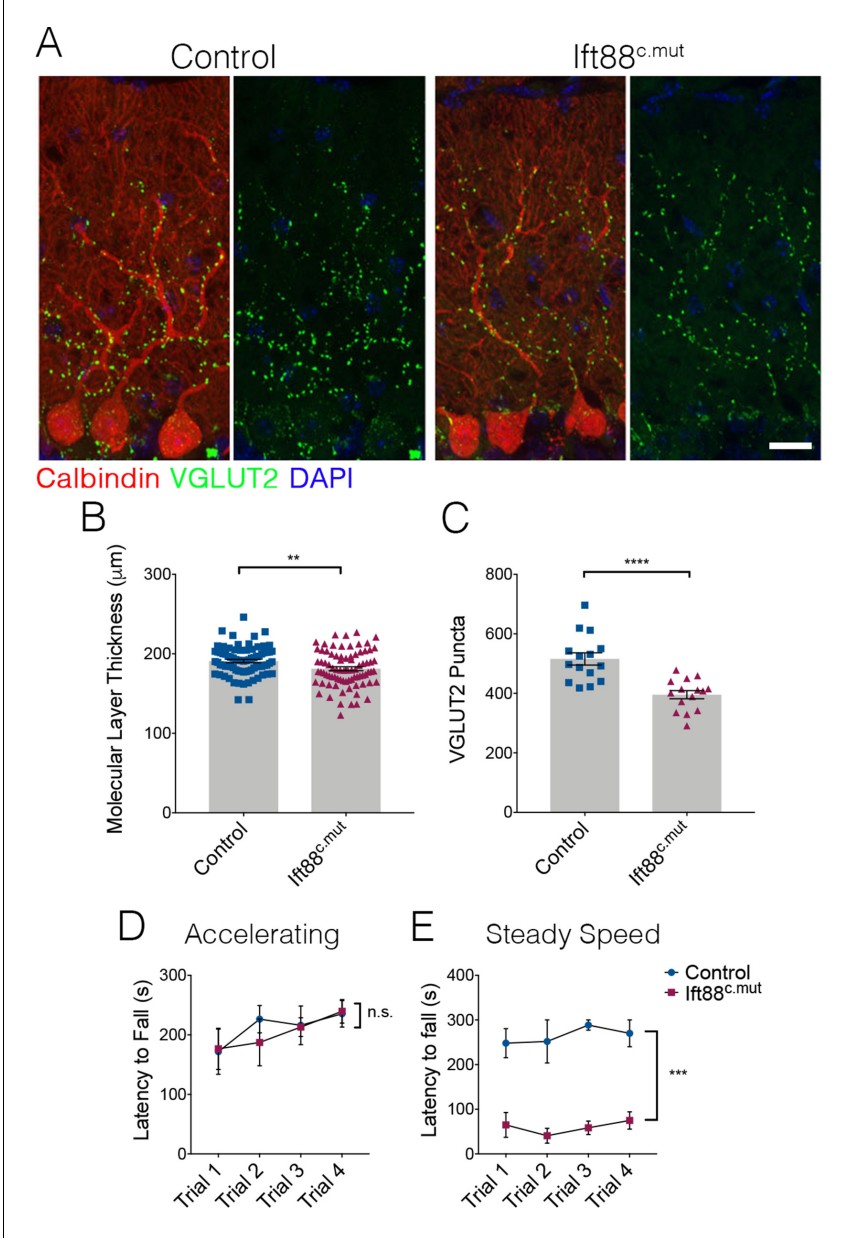

**Figure 7.** Loss of IFT88 recapitulates neurodegenerative phenotypes of Ttbk2[c.mut] animals. (**A**) Cerebellar tissue from Control and Ift88[c.mut] mice at 3 months after loss of *Ift88*, immunostained for Calbindin to label Purkinje cells (red) and VGLUT2 to show climbing fiber synapses (green). Ift88[c.mut] animals show a reduction in VGLUT2 positive synapses throughout the cerebellum 3 months after loss of IFT88. Scale bar = 50 µm. (**B**) Molecular layer length quantification of Ift88[c.mut] animals compared to littermate Controls. Each point represents one measurement, >75 measurements taken per genotype. n = 3 animals. p=0.0037 by unpaired student's t-test, error bars indicate SEM. (**C**) Quantification of loss of VGLUT2 synapses along PC dendrites in Ift88[c.mut] animals. Each point represents a field analyzed, with five images analyzed per animal, n = 3 animals. p<0.0001 by unpaired student's t-test, error bars indicate SEM. (**D, E**) Accelerating and steady speed rotarod performance test between Ift88[c.mut] and littermate Controls. Ift88[c.mut] animals do not have a significance difference in latency to fall time on the accelerating rotarod; however, the steady speed rotarod test showed a significantly shorter latency to fall time compared to Controls (a two-way ANOVA with Bonferroni's multiple comparison test was used for calculating significance. p=0.8343 for accelerating rotarod test, and p=0.0005 for steady speed. n = 6 animals for Control, n = 4 animals for Ift88[c.mut]).

concentrations through either ciliary loss or dysfunction could therefore result in perturbed signaling outputs to the cell bodies of these neurons. Provocatively, we found that AC3, a molecule important for the production of cAMP, is largely absent from the cilia that remain in the brains of Ift88[c.mut] animals, suggesting that its loss from the cilium may be a contributing signaling mechanism to the cellular changes in the cerebellum. Additionally, primary cilia play a well-known role as essential mediators of Hedgehog (HH) signaling. Sonic Hedgehog (SHH) is secreted by PCs during development well into adulthood (*Lewis et al., 2004*; *Traiffort et al., 1998*), although the precise role and functional significance of SHH within the adult cerebellum is largely unclear. Ultimately, it will be important to investigate the regulation and molecular composition of neuronal cilia in a comprehensive and unbiased manner.

Our observations also have interesting implications for regulation of cilium assembly and stability both generally, and in post-mitotic adult cells and neurons in particular. We found, for example, that

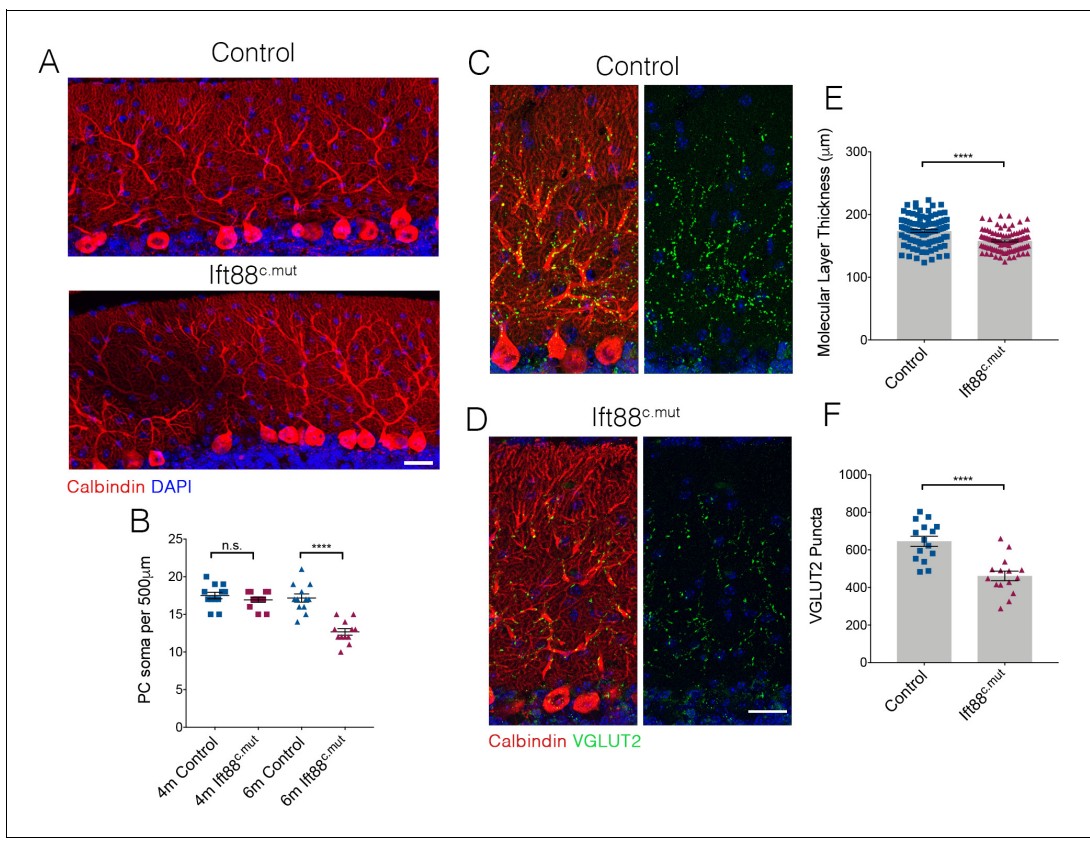

**Figure 8.** Loss of PCs occurs in Ift88[c.mut] mice by 6 months of age. (**A**) Cerebellar folia from Control and Ift88[c.mut] at 6 months of age, 5 months after loss of *Ift88*, stained for Calbindin (red) and DAPI (blue). 6-month-old Ift88[c.mut] mice begin to show PC gaps throughout cerebellar folia indicating loss of PC. Scale bar = 50 μm. (**B**) Quantification of number of PC soma per 500 μm stretch of folia on the primary fissure between Control and Ift88[c.mut] at 4 months and 6 months of age. 6-month-old Ift88[c.mut] show a reduced number of PC soma throughout the cerebellum (each point represents one measurement, 12 measurements were made per condition. n = 3 animals per condition. p<0.0001 by one-way ANOVA between all conditions, error bars indicate SEM). (**C, D**) 6-month-old cerebellar tissue from Control and Ift88[c.mut] mice immunostained for Calbindin to label PC (red) and VGLUT2 to show climbing fiber synapses (green) and DAPI (blue). Ift88[c.mut] animals show a reduction in VGLUT2 positive synapses throughout the cerebellum. Scale bar = 50 μm. (**E**) Quantification of molecular layer thickness between 6-month-old Control and Ift88[c.mut] animals. Ift88[c.mut] have shorter folia compared to littermate Controls (each point represents one measurement, >75 measurements taken per genotype. n = 3 animals. p<0.0001 by unpaired student's t-test, error bars indicate SEM). (**F**) Quantification of loss of VGLUT2 synapses along PC dendrites in 6-month-old Ift88[c.mut] animals. Ift88[c.mut] show a loss similar to the loss seen in 4-month-old Ift88[c.mut] animals (each point represents a field analyzed, with 5 images analyzed per animal, n = 3 animals. p<0.0001 by unpaired student's t-test, error bars indicate SEM).

most IFT88 protein is lost in *Ift88* conditional mutants as expected. However, the cilia persist on adult neurons within the brain in the absence of IFT88, suggesting that fully functional IFT is not required for these cilia. In contrast, in the absence of TTBK2, cilia are rapidly lost. This suggests that, in this context, TTBK2 could be playing a more central role than the IFT machinery in maintaining the stability of cilia. The exact mechanisms by which TTBK2 regulates the stability of cilia and the degree to which this role may be specifically important in neurons will be the subject of future investigations within our lab. In particular, the dynamics of cilium assembly and disassembly in post-mitotic cells in vivo have not been characterized. For example, a recent study found that the proteins that make up the basal body of adult neurons in the mouse are very long-lived whereas those of the ciliary axoneme turn over more quickly (*Arrojo E Drigo et al., 2019*). This might imply that the cilia of these adult neurons turn over at some interval, or simply that their protein components are replaced - a topic that merits further investigation.

In addition to being required for the biogenesis of cilia, TTBK2 also localizes to the + tips of microtubules, via interaction with the + end binding protein EB1 (*Jiang et al., 2012*). TTBK2 has also been shown to phosphorylate β-Tubulin as well as microtubule associated proteins TAU and MAP2 through in vitro assays (*Takahashi et al., 1995*; *Tomizawa et al., 2001*), pointing to roles for TTBK2 in regulation of the microtubule cytoskeleton beyond the cilium. In addition, TTBK2 and the closely related kinase TTBK1 both phosphorylate SV2A in vitro, a component of synaptic vesicles important for retrieval of the membrane trafficking protein Synaptotagmin one during the endocytosis of synaptic vesicles (*Zhang et al., 2015*). Although our data show that the *Ttbk2* mutant phenotypes strongly overlap with those of other ciliary genes, such as *Ift88*, we cannot exclude the possibility that other roles of TTBK2 specifically within the brain also contribute to the degenerative phenotypes. Importantly, these two possibilities are not mutually exclusive, and indeed, one exciting possibility is that TTBK2 is important for relaying signals from the cilium to the neuronal cell body.

In this work, we present evidence that loss or impaired function of TTBK2 within the brain results in degeneration of PCs largely because of the requirement for TTBK2 in mediating the assembly and stability of primary cilia. This points to ciliary dysfunction as being a major mechanism underlying the pathology of SCA11, which is caused by truncating mutations to *TTBK2* (*Houlden et al., 2007*) that act as dominant negatives (*Bowie et al., 2018*). In addition, our work raises the possibility that cilia play an important, largely unappreciated role in maintaining neuronal connectivity within the brain, and may also be required for the viability of some types of neurons. From a clinical perspective, our findings suggest that neurodegeneration, in addition to other neurological impairments with developmental origins, may emerge in some patients with ciliopathies such as Joubert and Bardet Biedl syndromes, particularly as patients age.

## Materials and methods

### Key resources table

| Reagent type (species) or resource | Designation | Source or reference | Identifiers | Additional information |
|---|---|---|---|---|
| Genetic reagent (*M. musculus*) | *Ttbk2*<sup>tm1a(EUCOMM)Hmgu</sup> | International Mouse Strain Resource | | |
| Genetic reagent (*M. musculus*) | *Actb*:FLPe | The Jackson Laboratory | #003800 | |
| Genetic reagent (*M. musculus*) | *Ift88*<sup>flox</sup> | The Jackson Laboratory | #022409 | |
| Genetic reagent (*M. musculus*) | *Ubc*-CreER | The Jackson Laboratory | #007001 | |
| Genetic reagent (*M. musculus*) | *Slc1a3*-CreER | The Jackson Laboratory | #012586 | |
| Genetic reagent (*M. musculus*) | *Pcp2*-Cre | The Jackson Laboratory | #010536 | |
| Chemical compound, drug | Tamoxifen | Sigma | T5648 | Working concentration: 20 mg/mL |

*Continued on next page*

*Continued*

| Reagent type (species) or resource | Designation | Source or reference | Identifiers | Additional information |
|---|---|---|---|---|
| Antibody | Mouse anti-ARL13B (monoclonal) | NeuroMabs | N295B/66 | 1:500 |
| Antibody | Rabbit anti-ARL13B (polyclonal) | Proteintech | 17711–1-AP | 1:500 |
| Antibody | Mouse anti-gamma-tubulin (monoclonal) | Sigma | T6557 | 1:1000 |
| Antibody | Rabbit anti-Calbindin D28K (monoclonal) | Cell signalling technologies | 13176S | 1:250 |
| Antibody | Guinea pig anti-VGLUT2 (polyclonal) | EMD Millipore | AB2251 | 1:2500 |
| Antibody | Rabbit anti-NeuN (monoclonal) | Abcam | Ab177487 | 1:1000 |
| Antibody | Rabbit anti-AC3 (polyclonal) | Santa Cruz | SC-588 | 1:10, discontinued |
| Antibody | Rabbit anti-AC3 (polyclonal) | Abeomics | 34–1003 | 1:100 |
| Antibody | Chicken anti-GFAP (polyclonal) | EMD Millipore | AB5541 | 1:500 |
| Antibody | Rabbit anti-IP3 (monoclonal) | Abcam | AB108517 | 1:200 |
| Antibody | Rabbit anti-FoxP2 (polyclonal) | Abcam | AB16046 | 1:400 |
| Antibody | Mouse anti-AT8 (monoclonal) | Thermo | MN1020 | 1:100 |
| Software | ImageJ | ImageJ: https://imagej.nih.gov/ij/ | | |
| Software | GraphPad Prism | GraphPad Prism: https://www.graphpad.com/scientific-software/prism/ | | |
| Commercial assay or kit | FD Rapid Golgistain Kit | FD Neurotechnologies | | |
| Commercial assay or kit | BSA Protein Assay Kit | Thermo | #23227 | |

## Ethics statement

The use and care of mice as described in this study was approved by the Institutional Animal Care and Use Committees of Duke University (Approval Number A218-17-09). All animal studies were performed in compliance with internationally accepted standards.

## Mouse strains

Ttbk2$^{c.mut}$ mice were produced by crossing *Ttbk2$^{tm1a(EUCOMM)Hmgu}$* mice to *ACTB:FLPe* (Jax stock #003800). The following mice were purchased from Jackson Laboratories: *Ift88$^{flox}$* (stock #022409), *Ubc-CreER* (stock #007001), *Slc1a3-CreER* (stock #012586) and *Pcp2-Cre* (Jax stock #010536). Slides used from *JNPL3(P301L)* mice were a gift from Dr. Carol Colton at Duke University.

## Genotyping

PCR genotyping was performed on all mice before experiments to confirm the presence of floxed alleles and Cre. *Ttbk2*-floxed allele, primers used: 5' ATACGGTTGAGATTCTTCTCCA, 3' AGGCTG TACTGTAACTCACAAT (WT band 978 bp, floxed band 1241 bp). *Ift88*-floxed allele, primers used:

5' GCCTCCTGTTTCTTGACAACAGTG, 3' GGTCCTAACAAGTAAGCCCAGTGTT (WT band 350 bp, floxed band 370 bp). Universal Cre (*Ubc-CreER*, *Pcp2-Cre*), primers used: 5' GATCTCCGGTA TTGAAACTCCAGC, 3' GCTAAACATGCTTCATCGTCGG (transgene band 650 bp).

## Tamoxifen preparation and injection

Tamoxifen powder (Sigma T5648) was dissolved in corn oil (Sigma C8267) to a desired concentration of 20 mg/mL. Mice were given five consecutive 100 µL intraperitoneal injections of 20 mg/mL tamoxifen starting at P21. Control mice were given corn oil vehicle only.

## Mouse dissections

To harvest tissues from adult mice, animals were deeply anesthetized with 12.5 mg/mL avertin and transcardially perfused with 10 mL of phosphate buffered saline (PBS) followed by 20 mL of 4% paraformaldehyde (PFA). Whole brains were dissected out and left to incubate for 24 h in 4% PFA at 4°C. For cryosectioning, tissue was cryoprotected in 30% sucrose overnight and embedded in Tissue Freezing Medium (General Data TFM-5). Cerebella were then cut sagittally down the middle, and embedded in Tissue Freezing Medium (General Data TFM-5). Tissue was sectioned at 20–30 µm thickness on a Leica Cyrostat (model CM3050S).

## Western blotting

Western blot methods were done as previously described (*Bouskila et al., 2011*) (*Bowie et al., 2018*). Briefly, for tissue which was being used to quantify levels of TTBK2, a buffer containing 50 mM Tris/HCl, pH 7.5, 1 mM EGTA, 1 mM EDTA, 1 mM sodium orthovanadate, 10 mM sodium-2-glycerophosphate, 50 mM sodium fluoride, 5 mM sodium pyrophosphate, 0.27 M sucrose, 1 mM benzamidine and 2 mM PMSF, supplemented with 0.5% NP-40 and 150 mM NaCl was used. For other tissue samples a buffer containing 10 mM Tris/Cl pH 7.5, 150 mM NaCl, 0.5 mM EDTA, 1% Triton, 1 mM protease inhibitors (Sigma #11836170001) and 25 mM β-glycerol phosphate (Sigma 50020) was used. Total protein concentration was determined using a BSA Protein Assay Kit (Thermo Fisher #23227). For western blots, 15 µg of protein lysate was used for detection.

## Cilia quantification

All quantification of cerebellar tissue was done using ImageJ software. Images taken for quantification of cilia abundance were 10 µm z-stacks taken at 63x in four distinct folia regions of the cerebellum, two rostral and two caudal (specifically, the outer edge of folia I/II, the internal zone between folia III and IV, the tip of folia VI, and the outer edge of folia IX on a sagittal section were imaged for cilia quantification). The Purkinje cell layer was placed into the middle of the image with equal distance above and below for quantification. Per animal, four sections were scored each and three animals were included in all quantifications. These cilia were the same population taken for cilia length measurements as well.

## Molecular layer thickness and VGLUT2 puncta quantification

For the molecular layer thickness, images were taken at 20x along the entirety of a primary fissure. A line was drawn from the base of the molecular layer to the pial surface, and a measurement was recorded. For this same line, the top of the line measurement was then brought down to the distal extent of the VGLUT2 synapse area, and a measurement recorded. For consistency, only the caudal side of the folia was measured. For the VGLUT2 puncta analysis, the 'Analyze Particles' function in ImageJ was used. Each image for the VGLUT2 puncta analysis was taken at 63x on the caudal side of the primary fissure, a 10 µm z-stack was made, and the image quantified. For the quantification, each stack was made into a black and white image, where the VGLUT2 puncta were black against a white background. Thresholding was performed, and the Analyze Particle function used. These measurements were routinely tested against user ROI counting to confirm accuracy. Four cerebellar slices were imaged per animal, and three animals were included in the analysis.

## Inferior olivary nuclei quantification

For the area measurements of the ION nuclei, the ION was identified by cells that were positive for both NeuN and Calbindin as well as location within ventral medulla in which these cells reside.

Images used for the NeuN area analysis were taken at 20x. A 10 µm z-stack image was made, and using the line tool, outlines were carefully drawn around the NeuN positive neuron and the area recorded. Per animal, over 150 cells were measured and three animals were included in the quantification.

## Glial fiber quantification

Glial fibers were assessed as previously described (*Furrer et al., 2011*). Briefly, a 100 µm horizontal line was drawn 50 µm below the pial surface of the primary fissure folia. Glial fibers which crossed this 100 µm were scored. Per animal, 36 measurements were made and three animals were included in the quantification.

## Immunostaining

The following antibodies and dilutions were used in this study: mouse anti-ARL13B (NeuroMabs N295B/66, 1:500), rabbit anti-ARL13B (gift from Tamara Caspary, 1:500, and Proteintech 17711–1-AP, 1:500), mouse anti-gamma-Tubulin (Sigma T6557, 1:1000), rabbit anti-Calbindin D28K (Cell Signaling Technologies 13176S, 1:250), guinea pig anti-Calbindin D28K (Synaptic Systems 214–004, 1:200), guinea pig anti-VGLUT2 (EMD Millipore AB2251, 1:2500), rabbit anti-NeuN (Abcam ab177487, 1:1000), DAPI (Sigma D9542, 1x), rabbit anti-AC3 (Santa Cruz SC-588, 1:10 - discontinued), rabbit anti-AC3 (Abeomics 34–1003, 1:100), chicken anti-GFAP (EMD Millipore AB5541, 1:500), rabbit anti-FoxP2 (Abcam, ab106046, 1:400), rabbit anti-IP3 (Abcam, ab108517, 1:200), and mouse anti-AT8 (Thermo Scientific MN1020, 1:100).

For immunostaining cerebellar tissue, sections were rinsed in 1xPBS to remove OCT and permeabilized in 0.2% PBS-T (PBS + 0.2% Triton X-100) for 10 min, and then rinsed 3 × 5 min in PBS before the blocking step. Blocking solution contained 5% serum, 1% BSA made up in 0.1% PBS-T, and sections were incubated at room temperature in blocking solution for 1 h. Primary antibodies were used at indicated dilutions and incubated at 4°C overnight. Following primary antibody incubation, slides were rinsed 3 × 5 min in 1xPBS and secondary antibodies were used to detect epitopes. All secondary antibodies were supplied from Life Technologies. Secondary antibodies incubated for 1–3 h at room temperature. Following secondary antibody incubation, slides were rinsed 3 × 5 min in 1xPBS and mounted with either ProLong Gold antifade reagent (Invitrogen P23930).

## Golgi staining

For Golgi staining of cerebellar tissue, FD Rapid Golgistain kit (FD Neurotechnologies) was used, according to manufacturer's instructions. 100 µm sections were made after staining. Imaging of Golgi stained PCs was completed using a Zeiss Axio Imager with a 40x objective. Z-stacks containing entire PCs were used for quantification, and proximal dendrites were chosen that did not intersect with other dendrites for clear quantification. Using ImageJ software, spines were counted per length of measurement and reported as number of spines per micron.

## Behavioral testing

A rotarod performance test was completed with help from the Duke University Mouse Behavioral and Neuroendocrine Core Facility. Testers were blind to mouse genotype before beginning any experiments. The accelerating rotarod testing was performed the day before steady state rotarod testing. All accelerating tests were conducted at a speed that increased from 4 RPM to 40 RPM over 5 min. All steady speed tests were conducted at 32 RPM. Four trials were conducted per test. A trial was stopped after 300 s maximum time had elapsed for mice that did not fall off the rotarod during testing. Mice were aborted from the trial run if they held onto the rotarod for three full rotations. Mice were given 30 min between trials to rest, and four trials were completed per test.

## Statistics

Statistical analyses, p-values, and experimental numbers for all experiments are outlined in respective figure legends. Analyses were performed using Graph Pad Prism 8.

## Acknowledgements

We are grateful to William Wetzel, Ramona Rodriguiz, and other staff of the Duke University Mouse Behavior and Neuroendocrine Core Facility for assistance with behavioral testing. We thank Drs. Don Fox and Anne West as well as members of the Goetz lab for helpful comments on the manuscript. This work was supported by grants from NIH/NICHD (R00HD076444) and the National Ataxia Foundation (Young Investigator Award in SCA) to SCG.

## Additional information

### Funding

| Funder | Grant reference number | Author |
|---|---|---|
| National Institutes of Health | R00 HD076444 | Sarah C Goetz |
| National Ataxia Foundation | Young Investigator award | Sarah C Goetz |

The funders had no role in study design, data collection and interpretation, or the decision to submit the work for publication.

### Author contributions

Emily Bowie, Conceptualization, Investigation, Methodology; Sarah C Goetz, Conceptualization, Resources, Formal analysis, Supervision, Investigation, Methodology, Project administration

### Author ORCIDs

Emily Bowie (iD) https://orcid.org/0000-0002-5694-6044
Sarah C Goetz (iD) https://orcid.org/0000-0001-9705-6390

### Ethics

Animal experimentation: This study was performed in strict accordance with the recommendations in the Guide for the Care and Use of Laboratory Animals of the National Institutes of Health. All of the animals were handled according to approved institutional animal care and use committee (IACUC) protocols (Protocol #A218-17-09) of Duke University. Every effort was made to minimize animal suffering.

### Decision letter and Author response

Decision letter https://doi.org/10.7554/eLife.51166.sa1
Author response https://doi.org/10.7554/eLife.51166.sa2

## Additional files

### Supplementary files

• Transparent reporting form

### Data availability

All data generated or analyzed during this study are included in the manuscript and supporting files.

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
