## [Decision Letter]

**Acceptance summary:**

The link between neurodegeneration and cilia has been mysterious, and this work provides some of the first solid advances in this area. A particular strength is the use of mouse genetics to address the cell types at the origin of SCA11. This work will be of interest to a broad swath of cerebellar researchers, cell biologists, developmental biologists, and clinicians interested in the mechanistic basis of neurodevelopmental and neurodegenerative diseases.

**Decision letter after peer review:**

Thank you for submitting your article "TTBK2 and primary cilia are essential for the connectivity and survival of cerebellar Purkinje neurons" for consideration by *eLife*. Your article has been reviewed by three peer reviewers, and the evaluation has been overseen by a Reviewing Editor and Marianne Bronner as the Senior Editor. The following individuals involved in review of your submission have agreed to reveal their identity: Roy V Sillitoe (Reviewer #1); Sylvie Schneider-Maunoury (Reviewer #2); Brian Kraemer (Reviewer #3).

The reviewers have discussed the reviews with one another and the Reviewing Editor has drafted this decision to help you prepare a revised submission.

The manuscript by Bowie and Goetz addresses the role of *Ttbk2* and cilia in the adult mouse cerebellum using the tamoxifen-inducible mutant (*Ttbk2*^c.mut^). *Ttbk2* (Tau tubulin kinase 2) is required for ciliogenesis in mice and TTBK2 dominant mutations in humans cause spinocerebellar ataxia type 11 (SCA11), a neural degenerative disease affecting the cerebellum. The study of *Ttbk2* function in the adult cerebellum is thus important to understand the origin of SCA11. The authors more particularly address the cell types at the origin of the disease as well as the involvement of cilia in adult cerebellum homeostasis.

Conditional ablation of *Ttbk2* using a global CreER as well as Purkinje cell specific Cre mice provided evidence that cilia are required for maintaining synapses in adult brain. Moreover, they report that Purkinje cells eventually die after genetic deletion. They confirm the phenotype by showing that loss of the ciliary trafficking gene *Ift88* results in a similar phenotype to the *Ttbk2* mice. They conclude that in the adult, cilia are required for maintaining neuronal function.

A key part of the manuscript is the use of a conditional KO of *Ttbk2* in the PCs (Ttbk2^PCP2^), since the global inducible KO *Ttbk2*^c.mut^ could lead to indirect effects on locomotor coordination and on cerebellum homeostasis. Strikingly, Ttbk2^PCP2^ animals shows similar, although milder defects than *Ttbk2*^c.mut^ animals.

This is an interesting study that uses a powerful and precise genetic strategy. The paper is generally well written. The topic is of interest to cerebellar researchers, cell biologists, developmental biologists, and clinicians interested in the mechanistic basis of neurodevelopmental and neurodegenerative diseases.

Major concerns:

1) The finding of different types of cilia in the cerebellum is interesting and raises a question as to whether different cilia present on different cell types. Unfortunately, different markers were used in the different mutants: ARL13B for *Ttbk2*^c.mut^ and ACIII and ARL13B for Ttbk2^PCP2^ (but surprisingly the quantification is not given for ARL13B +ACIII- cilia). Moreover, it would be very interesting to test for cilia loss in the PCs of Ttbk2^PCP2^ animals. Indeed, the milder VGLUT2 density and behavioural phenotypes found in Ttbk2^PCP2^ compared to *Ttbk2*^c.mut^ mice suggest that either cilia are not as affected in the PCP2 mutant or that other cell types are involved in the *Ttbk2*^c.mut^ phenotype.

2) The quantification of VGLUT2 puncta raises the question of the complexity and extent of the PC dendritic tree. Is the total number of VGLUT2 puncta reduced due to the reduction of the dendritic tree or is the density of puncta reduced on the dendrites? The question is important because it changes the interpretation of the role of cilia in PCs. It is important to better visualize the PC dendritic tree as the quantification of the thickness of the molecular layer is not sufficient.

3) The authors repeatedly make reference to the pathology of SCA11. In addition to exhibiting loss of cilia and PCs, SCA11 exhibits significant accumulation of pathological tau in brain regions outside the cerebellum. This key neuropathological feature of SCA11 is not addressed in the manuscript and should be mentioned in the Introduction and Discussion sections because pathological tau is a prominent feature of SCA11 and the other disease related phenotypes in humans are discussed extensively. It would be extremely significant from a disease modelling standpoint to assess whether or not TTBK2 conditional mice exhibit similar pathological tau accumulation outside the cerebellum vis-à-vis SCA11.

4) The main concern is that the analyses are based exclusively on the general architecture of the cerebellum. The cerebellum is heterogenous in its molecular, anatomical, and functional composition. It was unclear what regions of the cerebellum lose PCs. Degeneration often occurs in a pattern, and lobule X seems to have neuroprotective properties. For examples, refer to the SCA1 data from Zoghbi lab, and Sarna and Hawkes, 2003. Where within the cerebellum does the degeneration take place? Are all domains affected equally?

---

## [Author Response]

Major concerns:1) The finding of different types of cilia in the cerebellum is interesting and raises a question as to whether different cilia present on different cell types. Unfortunately, different markers were used in the different mutants: ARL13B for Ttbk2^c.mut^ and ACIII and ARL13B for Ttbk2^PCP2^ (but surprisingly the quantification is not given for ARL13B+ACIII- cilia). Moreover, it would be very interesting to test for cilia loss in the PCs of Ttbk2^PCP2^ animals. Indeed, the milder VGLUT2 density and behavioural phenotypes found in Ttbk2^PCP2^ compared to Ttbk2^c.mut^ mice suggest that either cilia are not as affected in the PCP2 mutant or that other cell types are involved in the Ttbk2^c.mut^ phenotype.

Unless otherwise noted, all cilia loss was quantified based on Arl13b immunofluorescence. Our initial analyses were all based on using Arl13b exclusively until we saw the result with AC3+ population changes in the Ift88^c.mut^ mice. AC3+ cilia are only analyzed with respect to the Ift88^c.mut ^ciliary phenotype, and we have simplified our discussion of the data for clarity. We have clarified this in the text and edited the graph axes for clarity throughout.

We have quantified the loss of cilia on PCs in P90 Ttbk2^PCP2^ mice. We saw a decrease in the number of PCs with primary cilia in Ttbk2^PCP2^ animals, and the results are reported in Figure 3—figure supplement 1 and added to the manuscript text.

2) The quantification of VGLUT2 puncta raises the question of the complexity and extent of the PC dendritic tree. Is the total number of VGLUT2 puncta reduced due to the reduction of the dendritic tree or is the density of puncta reduced on the dendrites? The question is important because it changes the interpretation of the role of cilia in PCs. It is important to better visualize the PC dendritic tree as the quantification of the thickness of the molecular layer is not sufficient.

We agree with the reviewers that a more thorough analysis of the PC dendrites is needed to better understand our VGLUT2 synapse loss data. We have performed Golgi staining on Ttbk2^c.mut^ brains to better visualize the morphology of individual PCs and to quantify spine density within the PCs. We report the results in Figure 1F and G as well as Figure 1—figure supplement 2, and have edited the text throughout the manuscript to include these findings. We did not find any difference of spine density between Controls and Ttbk2^c.mut^ PCs by counting amount of spines per unit length of the proximal dendrites. We chose to quantify the proximal dendrites as these are the set of dendrites with the VGLUT2+ associated synapses.

3) The authors repeatedly make reference to the pathology of SCA11. In addition to exhibiting loss of cilia and PCs, SCA11 exhibits significant accumulation of pathological tau in brain regions outside the cerebellum. This key neuropathological feature of SCA11 is not addressed in the manuscript and should be mentioned in the Introduction and Discussion sections because pathological tau is a prominent feature of SCA11 and the other disease related phenotypes in humans are discussed extensively. It would be extremely significant from a disease modelling standpoint to assess whether or not TTBK2 conditional mice exhibit similar pathological tau accumulation outside the cerebellum vis-à-vis SCA11.

At the reviewers’ suggestion, we have edited the manuscript to include a more thorough description of SCA11 pathology. We have performed immunostaining for AT8+ (pathologically phosphorylated) Tau in our 4 month and 6 month old Ttbk2^c.mut^ brains (both cortex and cerebellum) and do not see any pathological Tau accumulation in 4 month old Ttbk2^c.mut^, but do see a few cells positive for mild tau accumulation in the 6 month old *Ttbk2*^c.mut^ cortex (reported in Figure 4—figure supplement 2). As a positive control, we stained *JNPL3(P301L)* samples characterized by extensive Tau accumulation at the same time as our conditional mutants. We note that evidence for accumulation of pathological Tau playing a substantial role in SCA11 pathology is somewhat limited. The observation of Tau accumulation in the cerebral cortex and other brain regions in SCA11- affected individuals is based upon a single post mortem examination of a 77 year old individual.

4) The main concern is that the analyses are based exclusively on the general architecture of the cerebellum. The cerebellum is heterogenous in its molecular, anatomical, and functional composition. It was unclear what regions of the cerebellum lose PCs. Degeneration often occurs in a pattern, and lobule X seems to have neuroprotective properties. For examples, refer to the SCA1 data from Zoghbi lab, and Sarna and Hawkes, 2003. Where within the cerebellum does the degeneration take place? Are all domains affected equally?

We have quantified the regions of the cerebellum that lose PCs in the 6 month old

Ttbk2^c.mut^ animals and have reported the findings in Figure 4 as well as Figure 4—figure supplement 1 and updated the manuscript with this data. Previous quantifications reported in Figure 4 were based on a 500 μm length measurements taken throughout the total length of the primary fissure, and number of PCs counted along the primary fissure. We have added a folia quantification showing that gaps are seen in all folia of Ttbk2^c.mut^ cerebella, with the exception of folia X. Gaps are enriched in the inner folia IV-VII in Ttbk2^c.mut^ cerebella.